# Modulation of Mitochondrial Metabolic Parameters and Antioxidant Enzymes in Healthy and Glaucomatous Trabecular Meshwork Cells with Hybrid Small Molecule SA-2

**DOI:** 10.3390/ijms241411557

**Published:** 2023-07-17

**Authors:** Charles E. Amankwa, Olivia Young, Biddut DebNath, Sudershan R. Gondi, Rajiv Rangan, Dorette Z. Ellis, Gulab Zode, Dorota L. Stankowska, Suchismita Acharya

**Affiliations:** 1Department of Pharmacology and Neuroscience, University of North Texas Health Science Center, Fort Worth, TX 76107, USA; charlesamankwa@my.unthsc.edu (C.E.A.); oliviayoung@my.unthsc.edu (O.Y.); biddut.debnath@unthsc.edu (B.D.); gondisr@gmail.com (S.R.G.); rajivrangan@my.unthsc.edu (R.R.);; 2North Texas Eye Research Institute, University of North Texas Health Science Center, Fort Worth, TX 76107, USA; dorette.ellis@unthsc.edu; 3Department of Pharmaceutical Sciences, College of Pharmacy, University of North Texas Health Science Center, Fort Worth, TX 76107, USA; 4Department of Microbiology, Immunology and Genetics, University of North Texas Health Science Center, Fort Worth, TX 76107, USA

**Keywords:** trabecular meshwork, mitochondria, oxygen consumption rate, extracellular acidification rate, small molecules, antioxidants

## Abstract

Oxidative stress (OS)-induced mitochondrial damage is a risk factor for primary open-angle glaucoma (POAG). Mitochondria-targeted novel antioxidant therapies could unearth promising drug candidates for the management of POAG. Previously, our dual-acting hybrid molecule SA-2 with nitric oxide-donating and antioxidant activity reduced intraocular pressure and improved aqueous humor outflow in rodent eyes. Here, we examined the mechanistic role of SA-2 in trabecular meshwork (TM) cells in vitro and measured the activity of intracellular antioxidant enzymes during OS. Primary human TM cells isolated from normal (hNTM) or glaucomatous (hGTM) post-mortem donors and transformed glaucomatous TM cells (GTM-3) were used for in vitro assays. We examined the effect of SA-2 on oxygen consumption rate (OCR) and extracellular acidification rate (ECAR) in vitro using Seahorse Analyzer with or without the oxidant, tert-butyl hydroperoxide (TBHP) treatment. Concentrations of total antioxidant enzymes, catalase (CAT), malondialdehyde (MDA), and glutathione peroxidase (GPx) were measured. We observed significant protection of both hNTM and hGTM cells from TBHP-induced cell death by SA-2. Antioxidant enzymes were elevated in SA-2-treated cells compared to TBHP-treated cells. In addition, SA-2 demonstrated an increase in mitochondrial metabolic parameters. Altogether, SA-2 protected both normal and glaucomatous TM cells from OS via increasing mitochondrial energy parameters and the activity of antioxidant enzymes.

## 1. Introduction

Affecting nearly 60 million people worldwide, primary open-angle glaucoma (POAG), accounts for approximately 90% of all cases of glaucoma [1]. It is characterized by progressive multi-tissue damage, including the trabecular meshwork (TM), optic nerve head, retina, lateral geniculate nuclei, and the visual cortex [1,2,3,4]. Despite strides made toward understanding the pathophysiology of glaucoma, there remains no grand unifying theory underpinning the pathophysiology of the disease [1,5]. While physiological levels of reactive oxygen species (ROS) serve as important redox signaling molecules, supraphysiologic concentrations lead to reversible and irreversible damage to biomolecules, causing growth arrest and cell death.

Oxidative stress has been implicated in the pathogenesis of POAG, along with raised intraocular pressure (IOP) and visual field damage. ROS-mediated pathological features have been identified in the aqueous humor (AH) of glaucoma patients and several rodent models of chronic intraocular pressure (IOP) elevation [6,7,8]. In addition, human in vivo studies have shown a decrease in antioxidant enzyme levels in the AH of POAG patients compared to their age-matched controls [9]. Interestingly, oxidative DNA damage is significantly more abundant in the TM than in any other tissue in the anterior chamber. This phenomenon is inevitable because TM is the most sensitive tissue in the anterior chamber to oxidative damage [10,11]. Given its critical role in maintaining IOP homeostasis, TM integrity and functional loss is essential to glaucomatous disease progression.

Several lines of evidence suggest that pathophysiological alterations in the mitochondria are linked with decreased mitochondrial oxidative phosphorylation (OXPHOS), altered mitochondrial membrane permeability, and acceleration of apoptosis via the oxidation of macromolecules [6,7,12,13]. Abu-Amero et al. showed that lymphocytes isolated from peripheral blood among glaucoma patients had significantly lower mean mitochondrial respiratory activity compared with their controls (21% decrease, *p* < 0.001) [14].

Mitochondria play a central role in cellular processes, such as energy production in the form of ATP via oxidative phosphorylation, regulation of ROS production, and calcium homeostasis [15]. Under physiologic conditions, mitochondrial ROS production is intimately regulated. However, failure of the electron transport chain could lead to a one-electron reduction of the molecular oxygen-generating superoxide (O_2_^−^), which can be dismutated to hydrogen peroxide (H_2_O_2_) and molecular oxygen. While antioxidant enzymes such as catalase (CAT), superoxide dismutase (SOD), glutathione peroxidase (GPx), and glutathione reductase (GRe) are expressed in the TM and counter ROS via several mechanisms, mitochondrial damage persists among POAG patients. There remains a need for mitochondrial-targeted therapies that can promote or restore mitochondrial OXPHOS and glycolytic function as a protective mechanism for the TM and retina [6].

Our novel hybrid nitric oxide (NO) donor and antioxidant molecule, SA-2, has previously been shown to reduce IOP in three rodent models of ocular hypertension, exert superoxide dismutase (SOD) activity, as well as scavenge ROS in retinal ganglion cells (RGCs) and TM cells [16,17,18,19]. The present study investigated the change in mitochondrial bioenergetics and other antioxidant enzymes status in three cellular models: cultured human primary TM cells isolated from healthy and glaucomatous donor eyes and transformed human glaucomatous cell line. Our goal was to determine (1) the effect of SA-2 treatment on mitochondrial complex inhibitors as a function of changes in the oxygen consumption rate (OCR) and extracellular acidification rate (ECAR); (2) the effect of pro-oxidant tertbutyl hydroperoxide (TBHP) on the OCR and ECAR parameters, and if normal and glaucomatous TM cells respond to SA-2 treatment equally; and (3) the effect of SA-2 treatment on other antioxidant enzymes status, such as CAT and GPx in TM cells with or without TBHP treatment. The results of this study will provide insight into the underlying mechanisms of compound SA-2 in protecting the TM from oxidative damage.

## 2. Results

### 2.1. SA-2 Treatment Protects Normal and Glaucomatous TM Cells from TBHP-Induced Cell Death

Previously we have reported that SA-2 has high tolerability to cultured primary human TM cells collected from healthy donor eyes [17]. Since the TM cells used in these experiments were gathered from different human donor eyes, we again tested for cell viability and cytoprotection activity of SA-2 at different concentrations (1 µM, 10 µM, 100 µM, and 1000 µM). After 24 h treatment using the MTT assay, no cytotoxic activity was observed after treatment with SA-2. Treatment with TBHP (EC_50_ 300 µM) resulted in a nearly 41% decrease in hNTM cell viability (Figure 1b), and SA-2 demonstrated dose-dependent protection of hNTM cells from TBHP-induced oxidative stress. SA-2 at 10 µM, 100 µM, and 1000 µM concentrations significantly increased the viability of hNTM cells compared to the TBHP-treated group (85.3%, 97.3%, and 100%, respectively), as shown in Figure 1b. We tested SA-2 at 100 µM and 1000 µM in hGTM cells and observed no cytotoxicity. In addition, we observed significant cytoprotective activity in TBHP + SA-2 (100 µM)-treated cells (Figure 1c).

### 2.2. Effect of SA-2 Treatment on the OCR in Normal and Glaucomatous Human TM Cells

Prior to the start of the experiment, an initial dose and cell density titration was performed to determine the appropriate concentration of mitochondrial inhibitor carbonyl cyanide 4-(trifluoromethoxy) phenylhydrazone (FCCP) and optimal cell density for hNTM, hGTM, and GTM-3 cells. The goal was to identify the optimal FCCP concentration and cell density that generates the highest OCR parameters suitable for both primary hTM and GTM-3 cells. The OCR was assessed either with or without SA-2 (1 µM, 10 µM, 100 µM, and 1000 µM) or TBHP (150 µM) or both following a 24 h incubation period. Figure 2a,b show trace graph representations of mean OCR normalized to cell counts in each well. Comparisons of OCR after sequential injections of mitochondrial stressors (oligomycin, carbonyl cyanide 4-(trifluoromethoxy) phenylhydrazone (FCCP), and rotenone/antimycin A) and SA-2 showed a statistically significant increase in the mean OCR with SA-2-treated groups relative to the control group in both normal and glaucomatous TM cells.

An evaluation of mitochondrial function through OCR in hNTM cells showed a significant increase in all metabolic parameters in the SA-2-treated group at 100 µM or 1000 µM (Figure 2c–f) compared to the control group. We observed a significant increase (~75%) in basal respiration and ATP-linked respiration with SA-2-treated cells compared to the control (Figure 2c,d). Additionally, maximal respiration was significantly increased (75–100%) in the SA-2-treated groups compared to the control (Figure 2e). Approximately a 50% and 120% increase in spare respiratory capacity was seen in the 100 µM and 1000 µM SA-2-treated cells, respectively, when compared to the control group (Figure 2f). The trace graph representation of SA-2 in hGTM showed improved OCR parameters compared to untreated hGTM cells (Figure 2b). However, no statistical difference was observed in both basal and ATP-linked respiration for both concentrations of SA-2. Among the SA-2 (100 µM)-treated hGTM cells, maximal respiration was significantly improved by ~60%. As expected, spare respiratory capacity was considerably enhanced.

#### 2.2.1. Effect of SA-2 Treatment on ECAR in Normal and Glaucomatous Human TM Cells

As a byproduct of glycolysis, the release of protons into the extracellular milieu and protons generated from substrate oxidation with the export of CO_2_ represents the extracellular acidification rate (ECAR). Here, we observed that SA-2 at lower concentrations (1 µM and 10 µM) and significantly increased the ECAR in hNTM cells relative to the control cells (Figure 3a). Where we observed a significant (~5 fold) decrease in both basal and maximal OCR to ECAR ratio at the lower concentrations (1 µM and 10 µM), higher concentrations of SA-2 (100 µM and 1000 µM) have no significant effect compared to the control cells (Figure 3b). Similarly, in hGTM cells, a lower concentration of SA-2 (100 µM) improved the ECAR significantly (Figure 3c). This clearly shows that a lower concentration of SA-2 has a significant effect on the glycolytic pathway, driving the energy production in the mitochondria/cell, while higher concentrations of SA-2 possibly augment the actions of OXPHOS and/or ATP synthase pathways to increase mitochondrial bioenergetics—a phenomenon requiring further investigation.

#### 2.2.2. Effect of SA-2 Treatment on the Mean OCR and ECAR after TBHP Treatment

After 24 h incubation of hNTM and hGTM cells with pro-oxidant TBHP (150 µM), no change in the mean OCR was observed in hNTM cells (Figure 4a,b) whereas a significant reduction in mean OCR was observed in hGTM cells (Figure 4e,f) relative to the control cells. While co-treatment with 100 µM of SA-2 (TBHP + SA-2) significantly (*p* < 0.05) increased the mean OCR in hGTM mitochondria compared to the TBHP-only-treated group (Figure 4f), no effect was found in hNTM cells. A significant (*p* < 0.05) increase in basal respiration and ATP-linked respiration was observed after treatment with SA-2. To our utmost surprise, SA-2 treatment demonstrated differential responses in both hNTM and glaucomatous hTM cells. While TBHP treatment significantly decreased the ECAR in both hNTM (Figure 4c,d) and hGTM cells (Figure 4g,h), SA-2 treatment significantly rescued the glycolytic activity only in hNTM cells and showed no effect on hGTM cells. This phenomenon indicates that the mitoprotectant ability of SA-2 in normal and glaucomatous cells uses differential bioenergetic pathways/parameters in the presence of pro-oxidants such as TBHP. Our observation suggests preferential upregulation of glycolytic flux in hNTM cells and a rather significant augmentation of the OXPHOS system in hGTM cells. This selective mode of action of SA-2 needs further investigation.

### 2.3. SA-2 Is More Potent Than TEMPOL in Increasing the Maximal Respiration and Spare Capacity in GTM-3 Cells

Compared to a known SOD mimetic compound TEMPOL, we observed here that at 100 µM concentration, TEMPOL did not have any effect on OCR or ECAR in the mitochondria of transformed glaucomatous TM (GTM-3) cells, while the compound SA-2 significantly (~100%) increased both maximal respiration and spare respiratory capacity (Figure 5b) as well as the ECAR (Figure 5c). By comparison of OCR to ECAR ratios with TEMPOL and SA-2 treatment, we found that SA-2 treatment has a significantly higher effect on mitochondrial respiration and increased OCR compared to the control or TEMPOL-treated cells, respectively (Figure 5d). This confirms that SA-2 is more potent than TEMPOL in preventing mitochondrial dysfunction and maintaining normal energy production.

### 2.4. SA-2 Treatment Improves Antioxidant Status in the hNTM and hGTM Cells

One of the major, potentially therapeutic attributes of the hybrid compound SA-2 is its ability to upregulate several antioxidant enzymes and improve antioxidant defense mechanisms in ocular tissues. To corroborate this, we investigated the effect of SA-2 (100 µM) on antioxidant enzymes, including total antioxidant, CAT, and GPx activities after 24 h incubation. Figure 6a shows that treatment with SA-2 (100 µM) in hNTM cells increases total antioxidant activity (3-fold, *p* < 0.001) compared to the control cells. While treatment with TBHP in hNTM cells significantly increased the antioxidant concentration (~2.5-fold) compared to the untreated control, possibly as a compensatory mechanism to increased ROS. In hGTM cells, there is a significant decrease in total antioxidant concentration (Figure 6d). A combination treatment of SA-2 (100 µM) + TBHP significantly increased the total antioxidant concentration (4-fold) as shown in Figure 6a. Catalase and GPx activities significantly (2.0–2.5-fold) increased after SA-2 treatment compared to the control group in hNTM cells. TBHP treatment had no significant effect on either CAT or GPX activity. At the same time, TBHP + SA-2-treated cells demonstrated a significant increase in both enzyme activities with reference to the TBHP-only group (*p* < 0.05), as shown in Figure 6b,c. TBHP + SA-2 treatment indicates a (2.0–2.5-fold) increase in GPx and catalase activities in hGTM cells, as shown in Figure 6e,f.

### 2.5. SA-2 Treatment Decreases Lipid Peroxidation in hNTM Cells

Lipid peroxidation is an essential biological metric for measuring OS in cellular targets. As a byproduct of lipid peroxidation, the formation of malondialdehyde (MDA), a biomarker of OS and cell death, was measured in hNTM cells. Here, we observed a significant reduction in MDA when hNTM cells were treated alone with SA-2 (100 µM) or in combination with TBHP (Figure 7a). Similarly, a live cell analysis of lipid peroxidation in hNTM cells showed trends indicative of decreased lipid peroxidation when SA-2 was co-treated with TBHP for 24 h (Figure 7b,c).

## 3. Discussion

Oxidative stress is an underlying phenomenon in many neurodegenerative diseases. It is critical in regulating various biological processes, including cell death, antioxidant enzyme balance, and mitochondrial oxygen consumption. Although recent studies have shown the promising therapeutic potential of medicinal agents such as manganese porphyrin [20], astaxanthin [21], and other biological antioxidants in POAG, most of these agents lack specificity to mitochondria and do not fully address TM-related pathology in glaucoma. Here, we demonstrate the involvement of SA-2 in the mitochondrial oxygen consumption rate, extracellular acidification rate, and antioxidant enzyme status using both glaucomatous and healthy trabecular meshwork cells.

Residing in the iridocorneal angle, the TM tissue consists of highly specialized fenestrated beams of extracellular matrix (ECM) and endothelial-like cells adjacent to the Schlemm’s canal. While the TM is essential to maintain physiologic AH outflow resistance, this tissue is particularly vulnerable to toxicity attributable to high concentrations of ROS during oxidative stress [22]. Previously, we showed that the hybrid class of compound SA-2 containing a nitric oxide donor and SOD mimetic functionalities scavenges ROS, upregulates SOD, and improves RGC survival [16,18,23]. The compound SA-2 is a novel structural derivative of a previously known stable piperidine nitroxide TEMPOL (4-hydroxy-2,2,6,6-tetramethylpiperidin-1-oxyl). TEMPOL has shown robust antioxidant activity in various biological systems but at very high concentrations (>5 mM). Lindsey et al. demonstrated that a water-soluble oral derivative of TEMPOL (OT-440) protects against or delays early degenerative responses occurring in RGCs following optic nerve injury [24]; however, no report was found on its activity in TM cells. Our previous reports showed that treatment with SA-2 does not compromise the viability of ocular cells, including transformed human TM cells (NTM-5), 661 W photoreceptor cells, and primary normal hTM cells at 100 µM concentration. In addition, SA-2 (50 µM and 100 µM) moderately protected both 661 W photoreceptor cells and primary hTM cells against H_2_O_2_-induced oxidative insult [17,23]. Consistent with our previous findings, SA-2 demonstrated a lack of toxicity in both normal (hNTM) and glaucomatous (hGTM) cells and significantly increased the viability of THBP-induced decrease in TM cells.

Oxygen levels in the eye are tightly maintained within narrow limits by cellular metabolic activities, which are considered relatively low: <10 mmHg around the lens or ~13 mmHg (~2% O_2_) in the anterior chamber angle [10]. As expected, oxidative stress triggers excess mitochondrial ROS production, which may lead to impaired oxidative phosphorylation (OXPHOS), compromised tissue integrity, and apoptotic cell death [7,14,25,26,27,28]. Kubota et al. showed that under glaucomatous injury, the oxygen consumption rate and extracellular acidification rate of human lens epithelial cells (LECs) were reportedly lower than non-glaucoma LECs [29]. In addition, McElnea et al. reported elevated ROS production, impaired mitochondrial function, and elevated cytosolic Ca^2+^ in lamina cribrosa cells in the optic nerve head (ONH) from human glaucomatous patients [7]. Interestingly, limited bioenergetic therapies are aimed at optimizing mitochondrial function and OS in glaucoma and neurodegenerative diseases. Iodoacetic acid, an inhibitor of the glycolytic pathway, was reported to increase AH outflow; however, further investigation demonstrated that the mechanism of increased outflow was not due to the decrease in glycolysis or lactate concentration [30,31]. Moreover, comparative experimental studies involving mitochondrial function have only explored OCR in rabbit LECs, kidney epithelial cells (MDCK), and transformed TM cells, without investigating the mitochondrial function in primary hNTM cells [29].

To better understand the bioenergetics profile of primary normal hTM cells, we explored the changes in mitochondrial function pre- and post-treatment with SA-2 through the assessment of OXPHOS and glycolysis. A functional analysis of two major energy-producing pathways, OXPHOS and glycolysis, demonstrated significant improvement of OCR and ECAR after co-treatment with SA-2 in the absence of TBHP in hTM cells. This observation underscores the assertion that primary human TM cells primarily generate energy through oxidative phosphorylation and could preferentially compensate for a switch to glycolysis, given the hypoxic milieu in the anterior chamber. Therefore, impairment of glycolysis may ultimately affect OXPHOS since both pathways intricately work together to meet the cell’s energy demands [32]. The ratio of OCR to ECAR showed that the preference of hNTM cells for ATP generation is the OXPHOS pathway following treatment with SA-2 at >10 µM. Similarly, below 10 µM concentration, a compensatory mechanism via glycolysis is possibly activated. This novel finding suggests that by adjusting the concentration of SA-2, we can either manipulate, supplement, or control the generation of ATP levels in TM cells for improved viability and adequate function. This observation requires further investigation.

We observed notable increasing trends of mean OCR in the SA-2-treated group compared to the untreated control groups in glaucomatous TM cells. However, in contrast to hNTM, we only noticed insignificant increasing trends for basal respiration and ATP-linked respiration. As expected, spare respiratory capacity and maximal respiration were improved significantly after SA-2 treatment. This observation is attributable to the fact that mitochondrial metabolism is a highly dynamic process in the context of cellular function. Generally, depending on the cells’ metabolic needs, the OCR is tightly controlled by ATP demand. Hence, when required, mitochondrial respiration can abruptly increase to the maximal level to perfectly match cellular needs. When cells are subjected to stress, energy demand invariably increases, with more ATP required to maintain cellular functions. The difference between basal respiration and maximal respiration constitutes the mitochondrial spare respiratory capacity. Cells with large spare respiratory capacity can produce more ATP and overcome stress. In this case, we observed that SA-2 improved spare respiratory capacity, characterizing the TM’s mitochondrial capacity to meet additional energy requirements beyond the basal level in response to acute cellular stress. Mechanistically, the nitroxide component of SA-2 is a redox-cycling, membrane-permeable enhancer, which promotes superoxide dismutation at rates like the SOD enzyme.

The ROS scavenging activity and improvement of the antioxidant status of SA-2 alleviates stress and hence improves TM OCR parameters. Maximal respiration is primarily determined by oxidation and the supply of substrates, such as pyruvate, malate, and glutamate. TBHP treatment compromised both OCR and ECAR of the normal and glaucomatous cells. We observed a differential response of SA-2 (100 µM) to TBHP-treated normal and glaucomatous cells. While SA-2 treatment increased ECAR in normal TM cells following TBHP pre-treatment, an increase in OCR was observed in glaucomatous cells. The higher ECAR values in the SA-2-treated groups in hNTM cells compared to the control signifies higher levels of glycolysis as a possible compensatory mechanism for failing mitochondria after treatment with rotenone/antimycin A (complex-V inhibitor). The higher mean OCR values, including an increase in basal respiration and ATP-linked respiration in the SA-2-treated groups in hGTM cells, could be attributed to the fact that glaucomatous cells require higher metabolic energy for their protection by the OXPHOS pathway. Small molecules, such as meclizine (50 µM) and fingolimod phosphate (50 nM), showed similar trends, as exhibited by SA-2 (100 µM) in hNTM cells [33,34,35,36]. Unlike the canonical inhibitors of OXPHOS that directly target the respiratory chain, drug-like meclizine has exhibited slower kinetics to accumulate in the mitochondria, thus leading to the concomitant increase in ECAR. In agreement with our secondary dose-response screening assays, this suggests that SA-2 (100 µM) does not directly uncouple the mitochondrial OXPHOS machinery but rather under situations of dramatic increase in OS, SA-2 can induce metabolic shifts by increasing OCR to generate ATP for TM. Despite this finding, strong experimental evidence has shown that the oral administration of a known SOD mimetic compound, TEMPOL (300 mg/kg/d), to rats prevented the effect of tumor necrosis factor-alpha (TNFα)-induced increase in superoxide (O_2_^−^) production [37]. Moreover, Da Silva et al. reported improved mitochondrial biogenesis via increased OXPHOS. They showed that the improved OXPHOS was a result of increased levels of transcriptional regulation gene receptor PPARδ with concomitant increases in peroxisome proliferator-activated receptor gamma coactivator 1-alpha (PGC-1α) following treatment with TEMPOL [38]. Further investigation is needed to understand the differential protective mechanism of SA-2 in normal vs. glaucomatous cells. However, we do not exclude the possible activation of transcriptional factors, such as nuclear factor erythroid 2-related factor (Nrf2) and mitochondrial respiratory complex assembly proteins.

Under normal physiological states, ocular tissues possess intrinsic antioxidant enzymes to counteract the harmful effect of OS generated from normal metabolism. However, the eye is invariably exposed to radiation, atmospheric oxygen, and physical abrasion, thus creating an environment that supports the generation of elevated ROS.

On the other hand, glaucomatous TM cells have a compromised antioxidant enzyme status and are known to produce more ROS. SA-2 has been shown to scavenge ROS in TBHP-treated TM cells. In addition to scavenging ROS, regenerating endogenous antioxidants and repairing oxidative damage to TM may contribute to slowing down the progression of POAG. A direct comparison of the SOD mimetic compound TEMPOL with SA-2 showed no significant effect on the mean OCR at the 100 µM concentration. In contrast, we noticed a higher maximal respiration rate in SA-2-treated cells than in FCCP-treated control cells, similar to our observation in normal TM cells. SA-2 exerts a SOD mimetic effect, which dismutases peroxy (HOO^.^) and superoxide (O_2_^−^) radicals. The nitric oxide (NO) released from SA-2 is quantified to be in picomolar to nanomolar concentrations and possibly also act as a powerful antioxidant, preserving mitochondrial and cellular integrity during oxidative stress by neutralizing the iron load (Fe^2+^) in the Fenton reaction (Figure 8) and inhibiting the formation of hydroxyl (OH) radicals generated from superoxides and H_2_O_2_ [39]. In our earlier publications, we demonstrated that the NO released from SA-2 did not increase peroxynitrite radicals (ONOO^−^) in TM cells [18] or promote protein nitrotyrosylation in mouse retinal ganglion cells [16]. Additionally, since oxidative stress is essentially a function of mitochondrial ROS production and antioxidant systems, the presence of an active antioxidant system to counter overt oxidative damage is critical to the survival of TM cells. Previously, we have shown that SOD activity was significantly increased, and ROS significantly decreased after treatment with SA-2 to normal hTM cells [17]. Here we demonstrated that two other key antioxidant enzymes, GPx and CAT, were also significantly upregulated following SA-2 treatment, augmenting the redox cycle. TBHP treatment decreased cell viability, mitochondrial respiration parameters, catalase, and GPX activities. Reduced glutathione and catalase function is reported to interfere with the metabolism of oxidative intermediates and exacerbate the direct or indirect damaging effects of oxidative stress on biological tissues [5,9,40,41]. We confirmed here that the increase in GPx and CAT activity and scavenging/neutralizing the ROS are the mechanisms for SA-2’s cyto/mitoprotecting activity to maintain cellular homeostasis in TM cells. There is evidence that lipid peroxidation metabolites are involved in oxidative reactions in the eye and play roles in the pathogenesis of many ocular diseases, including POAG [42,43,44,45,46]. We measured malondialdehyde (MDA), a byproduct of lipid peroxidation in hNTM cells, and as expected, a significant decrease was observed post-treatment with SA-2.

## 4. Materials and Methods

### 4.1. Materials and Reagents

Oligomycin, rotenone, antimycin A, carbonyl cyanide 4-(trifluoromethoxy) phenylhydrazone (FCCP), and tert-butyl hydroperoxide (TBHP) were purchased from Sigma Aldrich (St. Louis, MO, USA). XF calibrant solution, XF Dulbecco’s Modified Eagle Medium (DMEM) medium, D-Glucose anhydrous, 100 mM pyruvate, 200 mM glutamine, Dulbecco’s modified Eagle’s medium (DMEM), Seahorse XF Cell Mitochondrial Stress Test Kit, and Extracellular Flux Assay Kits were all purchased from Agilent Technologies (Wood Creek, TX, USA). 3-(4,5-dimethylthiazol-2-yl)-2,5-diphenyltetrazolium Bromide (MTT) was purchased from Promega (Madison, WI, USA), trypsin-EDTA-0.05% from ThermoFisher Scientific (Merelbeke, Belgium), and all the reagents for the synthesis of SA-2 were purchased from Sigma Aldrich (St. Louis, MO, USA). Primary human trabecular meshwork (hTM) cells were isolated from post-mortem healthy (n = 3) or glaucomatous (n = 3) human donor eyes (Willed Body Program, UNTHSC, Fort Worth, TX, USA), characterized as previously described by dexamethasone-induction of myocilin and absence of VE-cadherin, and cultured for all in vitro studies [47,48]. The transformed glaucomatous trabecular meshwork (GTM-3) cell line used in our study was transformed from glaucomatous TM by transfecting primary TM cells with an origin-defective mutant of simian virus 40 [49]. These cells were generous gifts from Abbot F. Clark to Dorette Ellis (Alcon Laboratories, Fort Worth, TX, USA).

### 4.2. Synthesis of SA-2

The compound SA-2 is prepared from commercially available amine in three steps using a modified protocol, as shown in Figure 1.

#### 4.2.1. N-(Cyanomethyl)-N-(1-Hydroxy-2,2,6,6 Tetramethylpiperidin-4-yl) Free Radical 2

To a mixture of amine 1 (500 mg, 2.92 mmol, 1.0 eq) in acetonitrile (20 mL) was added successively 443 mg (3.21 mmol, 1.1 eq) of potassium carbonate and 385 mg (3.21 mmol, 1.1 eq) of bromo acetonitrile. The mixture was refluxed for 6 h, cooled to room temperature, and filtered, and the solids were washed with a 9:1 ratio of chloroform:methanol (90 mL:10 mL). The filtrate was concentrated to obtain 600 mg residual oil, which was purified on silica gel column chromatography using 10% methanol in ethyl acetate (EtOAc) to obtain 450 mg of product 2 in a 73.2% yield. ^1^H Nuclear Magnetic Resonance (NMR) (300 MHz, CDCl_3_-one drop of CD_3_OD): δ 4.61 (s, 2H), 4.47-4.20 (m, 2H), 3.57–3.51 (3, 2H), 1.81–1.56 (m, 6H), 1.36–1.26 (m, 6H). ^13^C-NMR (75 MHz, DMSO-d_6_): δ 119.48 (-CN), 67.27 (-C(CH_3_)_2_), 64.47 (-CH-N=), 46.35 -CH_2_-CH-N=), 36.70 (CH_2_-CN), 25.52 (-C(CH_3_)_2_), 24.31 (-C(CH_3_)_2_). TOF-Mass: C_11_H_20_N_3_O (M + H): 211.1679.

#### 4.2.2. N-(Cyanomethyl)-N-(1-Hydroxy-2,2,6,6-Tetramethylpiperidin-4-yl) Nitrous Amide 3

To a pre-cooled solution of 2.85 mmol (600 mg) nitrile 2 in acidic H_2_O (10 mL containing 3.2 mL of 1 N HCl) at 0 °C was added 220 mg of NaNO_2_, and the reaction mixture was stirred at 0 °C for 1 h to form a white precipitate. The pale-yellow solid was filtered, washed with n-hexane, and dried under vacuum to obtain 600 mg of pale-white solid 3 in 88% yield. ^1^H NMR (300 MHz, DMSO-d_6_): δ 11.64 s, 1H, OH), 5.05–4.97 (m, 1H), 4.68 (s, 2H), 2.69–2.61 (t, 2H, J = 13.3 Hz), 2.35–2.31 (d, 2H, J = 13.2 Hz), 1.54 (s, 6H), 1.43 (s, 6H). ^13^C-NMR (75 MHz, DMSO-d_6_): δ 114.29 (-CN), 67.60 (-C(CH_3_)_2_), 54.38 (-CH-N=), 40.19 (-CH_2_-CH-N=), 32.47 (CH_2_-CN), 27.73 (-C(CH_3_)_2_), 20.20 (-C(CH_3_)_2_). TOF-Mass: C_11_H_20_N_4_O_2_ (M + H): 241.1552. FT-IR (neat, λ cm^−1^): 2971 C-H (s), 2721 N=O (s), 2150 CN (s), 2075 C-N-N (s), 1510 N=O (b), 1452 C-H (b), 1385 (C-H) gem-alkyl (b), 1342 C-N (b).

#### 4.2.3. 5-Amino-3-(1-Hydroxy-2,2,6,6-Tetramethylpiperidin-4-yl)-1,2,3-Oxadiazol-3-Ium Chloride (SA-2)

The nitroso compound 3 (250 mg, 1.35 mmol) was suspended in methanol (5 mL), 5 mL of 1 M methanolic hydrochloric acid (HCl) was added, and the reaction mixture was stirred overnight at room temperature to form a clear solution. The solvent was removed, and the crude mixture was dissolved with 1 mL of methanol, followed by the addition of 40 mL of diethyl ether to precipitate the white solid. The solids were filtered, washed with diethyl ether (20 mL), and dried under a vacuum to obtain 165 mg of white powder in 55%. ^1^H NMR (300 MHz, D_2_O): δ 7.70 (s, 1H), 5.50–5.43 (m, 1H), 2.68–2.63 (d, 2H, J = 13.2 Hz), 2.42–2.37 (t, 2H, J = 13.1 Hz), 1.47 (s, 6H), 1.44 (s, 6H). 13C-NMR (75 MHz, D_2_O): δ 169.50 (O-C-NH), 102.07 (N-C=), 67.94 (-C(CH_3_)_2_), 56.3 (-CH-N=), 40.2 (-CH_2_-CH-N=), 26.89 (-C(CH_3_)_2_), 18.97 (-C(CH_3_)_2_). TOF-Mass: C_11_H_21_N_4_O_2_ (M+H): 242.1760. FT-IR (neat, λ cm^−1^): 3550 OH (b), 2989C-H (s), 2665 N=O (s), 2388 NCO (s), 1678 CON (s), 1572 N=O (b), 1473 C-H (b), 1388 (C-H) gem-alkyl (b), 1311 C-N (b).

### 4.3. Culture of Primary hNTM and hGTM Cells

Post-mortem eyes obtained within 24 h were collected and dissected for TM cell isolation. Following isolation and characterization, hTM cells were cultured in low-glucose DMEM (Gibco^®^, St. Louis, MO, USA) containing 10% fetal bovine serum, 1% glutamine, and 1% penicillin–streptomycin (100 U/mL penicillin and 100 ng/mL streptomycin) for functional assays. Primary TM cells within a passage range of 6–8 were grown to confluence in T25 flasks, then transferred to either a 96-well, 24-well, or 6-well plate for functional assays depending on the type of study. Similarly, GTM-3 cells were cultured in low-glucose DMEM (Gibco^®^, St. Louis, MO, USA), 10% fetal bovine serum, 1% glutamine, and 1% penicillin–streptomycin (100 U/mL penicillin and 100 ng/mL streptomycin) to achieve confluency and later serum-deprived for functional assays. GTM-3 cells within a passage range of 18–20 were cultured to confluency in T75 flasks, then passaged to either a 96-well or 6-well plate for in vitro studies. The optimal cell number and time protocol for each assay were experimentally determined.

### 4.4. In Vitro Cell Viability Assay

Cell viability was assessed using a colorimetric 3-(4,5-dimethylthiazol-2-yl)-2,5-diphenyltetrazolium Bromide (MTT) reagent (CellTiter96^®^ Aqueous One Solution Cell Proliferation Assay, Promega Madison, WI, USA). Primary human trabecular meshwork (hTM) cells (30,000 cells/well/200 µL) were seeded in 96-well plates to confluency. The cells were serum-deprived for 24 h and were exposed to different concentrations of SA-2 (1 µM, 10 µM, 100 µM, and 1000 µM). After 24 h of incubation at 37 °C, cell viability was measured using an MTT assay (CellTiter 96^®^ AQueous One Solution Cell Proliferation Assay, Promega Madison, WI, USA) following the manufacturer’s protocol. Experiments were repeated twice with three technical replicates, and the percentage of viable cells was determined by normalizing to the untreated control group.

### 4.5. In Vitro Cytoprotection Assay

In parallel, primary hNTM (n = 3), hGTM (n = 3), and GTM-3 cells (30,000 cells/well) were seeded in 96-well plates to confluency. The cells were starved for 24 h with serum- free media. Cells were treated with 300 µM of t-butyl hydrogen peroxide (TBHP) for 30 min, followed by co-treatment with SA-2 at different concentrations (1 µM, 10 µM, 100 µM, and 1000 µM). Cell viability was measured using an MTT assay. Experiments were repeated twice with three technical replicates, and the percentage of viable cells was determined by normalizing to the untreated control group.

### 4.6. Measurement of Oxygen Consumption Rate and Extracellular Acidification Rate

Analysis of glycolytic function and OCR was performed on live primary hTM and GTM-3 cells independently using the Agilent Seahorse XFe24 Analyzer (Agilent Seahorse Bioscience, Santa Clara, CA, USA). In brief, a day before the assay, hNTM and hGTM cells (passages 6–8) and GTM-3 cells (passages 18–20) were seeded at 60,000 cells per well (optimized cells density) in a poly-D-lysine coated seahorse XFe24 cell culture microplate. The monolayered cells were allowed to adhere to the plate for 24 h in a 37 °C humidified incubator with 5% CO_2_. After achieving ~95% confluence, cells were serum-deprived for 24 h and treated with SA-2 at 1 µM, 10 µM, 100 µM, and 1000 µM with or without TBHP (150 µM) concentrations. After 24 h incubation, cells were washed with XF-based medium DMEM supplemented with 10 mM glucose, 2 mM sodium pyruvate, and 2 mM glutamine (adjusted to pH_7.4_) and maintained in 450 µL·well^−1^ at 37 °C in a non-CO_2_ incubator for 1 h to allow for pre-equilibration with the XF Assay Medium.

Mitochondrial OCR and ECAR were analyzed following sequential injections of modulators, including oligomycin (1.5 μM; port A)—an ATP synthase inhibitor; carbonyl-cyanide-4-(trifluoromethoxy) phenyhydrazone (Sigma-Aldrich, Steinheim, Germany), (FCCP; 0.5 µM; port B)—a mitochondrial oxidative phosphorylation uncoupler; and rotenone (Sigma-Aldrich, Steinheim, Germany) together with antimycin A (Sigma-Aldrich, Steinheim, Germany), (0.5 μM and 0.5 μM; port C)—complex I and complex III inhibitors. Using a 24 min injection interval for four cycles, mitochondrial oxidative phosphorylation parameters, such as basal respiration, ATP-linked respiration, maximum respiratory capacity, as well as extracellular acidification rate (ECAR), were measured and normalized to the number of cells per well after the completion of the assay. The cells were carefully washed with PBS and fixed in 2% paraformaldehyde (PFA) to normalize. Afterward, cells were washed with tris buffered saline, stained with DAPI (1:1000), and washed twice with PBS for nuclear cell count. Cells were imaged using a BioTek Cytation 5 microplate reader (BioTek Instruments Inc., Highland Park, Winooski, VT, USA) and counted manually by two persons in a masked manner. The metabolic parameters and OCR to ECAR ratio were calculated using a previously published method [50]. Three independent experiments with three wells per group were performed and analyzed using an Agilent XFe24 seahorse analyzer (Agilent Technologies, Lexington, MA, USA).

### 4.7. Antioxidant Enzyme Assay

Cultured primary hNTM or hGTM cells from post-mortem human donor eyes (passage 6–10) were seeded (400,000 cells/well/2 mL) in a 6-well plate to achieve approximately 95% confluency. The cells were treated with SA-2 (100 µM) only, SA-2 (100 µM) + TBHP (150 µM), TBHP (150 µM) only, as well as the untreated group as a negative control. Cell lysates were extracted on ice with M-PER™ Mammalian protein extraction reagent (Thermo Scientific, Rockford, IL, USA) and homogenized in cold buffer (50 mM Tris-HCl, Ph 7.7, 5 mM EDTA, and 1 m MDTT). The amount of total protein was quantified using the bicinchoninic acid (BCA) assay (Thermo Scientific, Rockford, IL, USA) and used for total antioxidant enzymes, catalase, and glutathione peroxidase assay, respectively, following Cayman assay protocols.

#### 4.7.1. Total Antioxidant Assay

To measure the total antioxidant concentration in hNTM or hGTM cell lysates, Cayman’s Antioxidant kit (catalog # 709001, Cayman Chemicals, Ann Arbor, MI, USA) was used. The assay relies on the ability of sample antioxidants to inhibit the oxidation of ABTS^®^ (2,2′-Azino-di-[3-ethylbenzthiazoline sulphonate]) to ABTS^®.+^ relative to Trolox (a water-soluble tocopherol analog) standard. Absorbance is measured spectrophotometrically at 750 or 405 nm.

#### 4.7.2. Catalase Assay

The method for determining catalase (CAT) activity is based on the peroxidative function of CAT. This is based on the reaction of methanol in the presence of an optimal H_2_O_2_ concentration. The formaldehyde produced is measured colorimetrically with 4-amino-3-hydrazino-5-mercapto-1,2,4-triazole (Purpald), the concentration of which is measured using spectrophotometry at 540 nm. One unit of CAT activity is defined as the number of enzymes causing the formation of 1 nM of formaldehyde per minute (nmol/min/mL). Cell lysates obtained from SA-2-treated hNTM or hGTM cells were used for the assay following the manufacturer’s protocol (catalog # 707002, Cayman Chemicals, Ann Arbor, MI, USA).

#### 4.7.3. Glutathione Peroxidase Assay

GPx activity is measured indirectly via a coupling reaction with glutathione reductase (GRe). The oxidized glutathione is reduced by GRe and NADPH. In this reaction, NADPH oxidation to NADP+ is accompanied by a decrease in absorbance at 340 nm, which is proportional to the GPx activity in the sample. Cell lysates obtained from SA-2-treated hTM and hGTM cells were used for the assay following the manufacturer’s protocol (catalog # 703102, Cayman Chemicals, Ann Arbor, MI, USA).

#### 4.7.4. Lipid Peroxidation Assay

Lipid peroxidation was assayed using colorimetric assays and live cells Image-iT lipid peroxidation reagent. For the colorimetric assay, the Lipid Peroxidation assay kit (ab233471, Abcam, Cambridge, UK) measures the reaction between a chromogenic reagent and malondialdehyde (MDA), the end product from peroxidation of polyunsaturated fats. Cell lysates obtained from SA-2-treated hNTM cells were collected and used for the assay following the manufacturer’s protocol. A standard curve was generated, and the sample’s MDA concentration was expressed as nmol per mg protein ± SEM.

#### 4.7.5. Immunocytochemistry-Staining for Lipid Peroxidation

For the live imaging of lipid peroxidation, human normal TM (hNTM) cells were plated on 35 mm glass-bottom dishes (MatTek, Ashland, MA, USA) and stained with a 10 µM Lipid Peroxidation Sensor for 30 mins in a complete growth medium at 37 °C. For TBHP treatment, the cells were pre-treated with 150 µM TBHP for 30 min. After 30 min incubation, the cells were then treated with a vehicle (only media) or 100 µM SA-2 or 150 µM tert-butyl hydroperoxide (TBHP) for 2 h at 37 °C. Using the Lipid Peroxidation assay kit (C10445, Invitrogen, Carlsbad, CA, USA), cells were stained with BODIPY™ 581/591 C11 reagent and Hoechst 33342 during the last 30 min of compound incubation. The cells were then washed 3 times with phosphate-buffered saline (PBS) and then imaged on a Zeiss confocal microscope using a 40× objective using filters for Hoechst, FITC, and Texas Red channels.

### 4.8. Statistics

Data were analyzed using one-way analysis of variance (ANOVA) with repeated measures and a multiple comparison Dunnett’s test or two-way ANOVA with Tukey’s multiple comparison test. Analyses were performed using GraphPad Prism 9 (GraphPad Software, San Diego, CA, USA) with a required significance level of *p* < 0.05. All data are represented as the mean of three experiments ± SEM. Three strains of hNTM and hGTM were used for the experiments.

## 5. Conclusions

In summary, the small hybrid molecule SA-2 with SOD mimetic and NO-donating activities previously evaluated in rodent models of glaucoma demonstrated significant protection of human primary TM cells from oxidative stress-induced cell death. The compound SA-2 exhibited promising protection against oxidative stress-induced damage in both healthy and glaucomatous human TM cells in vitro. An evaluation of the mitochondria protective activity of SA-2 in TBHP-stressed TM cells demonstrated improved mitochondrial oxidative phosphorylation and glycolysis parameters. Further investigation is needed to better understand SA-2’s selective mitochondria protective efficacy in glaucomatous vs. normal TM cells and how these in vitro findings would translate to in vivo models.

## Data Availability

The data presented in this study are available on request from the corresponding author.

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
