# Peer review of "Modulation of Mitochondrial Metabolic Parameters and Antioxidant Enzymes in Healthy and Glaucomatous Trabecular Meshwork Cells with Hybrid Small Molecule SA-2"

_ijms, 2023, doi:10.3390/ijms241411557_

Round 1

Reviewer 1 Report

Are differences between hNTM and hGTM cells observed unique to this pair, or are they also seen with additional isolates from different donors? 

Different doses of SA-2 appear to act on different aspect of mitochondrial functionality.  Pls juxtapose these mechanistic results with the previously published therapeutic studies using various dose of drugs and speculate what pathways are most beneficial.  

Reviewer 2 Report

The study showed a protective role of SA-2 molecule from oxidative stress in human TM cells. The results are encouraging and may build a basis for further research to develop new treatment strategies. I'd like to congratulate the authors for their great achievement.

Author Response

We thank the reviewer for a thorough and professional assessment of the manuscript. We greatly appreciate the reviewer's encouraging comments.

Reviewer 3 Report

The manuscript entitled “Modulation of Mitochondrial Metabolic Parameters and Antioxidant Enzymes in Healthy and Glaucomatous Trabecular Meshwork Cells with Hybrid Small Molecule SA-2” is significant in this field of interest. The manuscript is well-structured with enough data. However, this manuscript has minor issues needed to be addressed. Thus I recommend this manuscript for minor revision.

Please ensure that the entire manuscript is thoroughly reviewed for typographical errors.

eg: for in-vitro as-, SA-2 (100µM) +TBHP compared, respectively (Fig.8a). and proliferation and 612 survival. .

Please check line numbers 98 and 110.

Avoid using abbreviations in their first occurrence.

Eg: Methanol in EtOAc to obtain (line no: 104); represents the % of viable cells after

Figures 3a and b, Figures 5a, c, e, and g, and Figure 6a require statistical analysis.

If there is no significant difference in values for Figures 3g and h, please mention it in a single line and explain the meaning of "ns" in the figure legend.

Please review Figure 3e and Figure 7C to verify the presence of the statistical indication mark "*", if applicable.

Please expand the abbreviation "FCCP" (0.5µM) in line number 312.

Lastly, check the y-axis unit in Figure 8a.
